# Sexual Orientation and the Incidence of COVID-19: Evidence from Understanding Society in the UK Longitudinal Household Study

**DOI:** 10.3390/healthcare9080937

**Published:** 2021-07-26

**Authors:** Cara L. Booker, Catherine Meads

**Affiliations:** 1Institute for Social and Economic Research, University of Essex, Essex CO4 3SQ, UK; 2Faculty of Health, Education. Medicine and Social Care, Cambridge Campus, Anglia Ruskin University, Cambridge CB1 1PT, UK; catherine.meads@aru.ac.uk

**Keywords:** sexual orientation, COVID-19, UKHLS, longitudinal study, health inequalities

## Abstract

COVID-19 infection rates and severity are worse in marginalised groups, although, for sexual and gender minorities, there are no data on infections, hospitalisations or deaths, but there may be worse rates. This study uses information from Understanding Society: The UK Household Longitudinal Study (UKHLS) to derive COVID-19 symptoms and positive tests by sexual orientation. Data came from all seven UKHLS COVID-19 survey waves in 2020 and 2021, and sexual orientation in main UKHLS waves 3 and 9. Numbers ranged from 17,800 to 12,000. Covariates in the regression models were gender, age, highest educational qualification, ethnicity, diagnosed medical condition, and key worker status. Compared to heterosexual individuals, more sexual minorities experienced symptoms, and bisexual individuals reported a greater number of symptoms. Gays and lesbians were no more or less likely to have been tested, but a larger proportion of bisexual individuals were tested. Regression models showed that differences mostly disappeared when other characteristics were considered. A small sample size means that principal questions remain, so health inequalities have been largely unnoticed and therefore not addressed. Suitable action should be taken to minimise their future risks. Why sexual and gender minorities have been omitted needs to be explored, and action needs to be taken to ensure this does not happen again.

## 1. Introduction

COVID-19 infection rates and severity have been worse in many societal groups who already face disadvantage and discrimination, particularly in people who have socioeconomic deprivation, who are from Black and minority ethnic backgrounds, and who are older people. The pandemic has exacerbated existing health inequalities [1,2], but there is a noticeable lack of information on sexual and gender minorities in these reports. Sexual minority status can be recorded through identity (lesbian, gay, bisexual), sexual behaviour (men who have sex with men (MSM), women who have sex with women (WSW), etc., or by cohabitation, civil partnership, or same-sex marriage). Gender minority status can be recorded through the presence of gender recognition certificates (but only around 12% of trans people have these) [3], or a self-report of whether their current gender is the same as that assigned at birth.

In sexual and gender minorities, the COVID-19 pandemic has worsened mental health and wellbeing, health behaviours, safety, social connectedness, and access to routine healthcare [4]. However, nothing has been published so far on the rates of infection, symptom severity, hospitalisations, intensive care unit admissions, or deaths from COVID-19 in UK sexual and gender minority populations. There are several reasons why there might be differential rates. For gay and bisexual men, there was an increased risk of having contact with a romantic/sexual partner outside of their household during the first UK lockdown [5], which may result in higher rates of COVID-19 infections. There are also higher rates of ill mental health [6], poorer wellbeing [7], and smoking [8] compared to heterosexual men. For lesbians and bisexual women, there are higher rates of obesity [9], asthma [10], and smoking [8]. In trans women, there are higher rates of mental ill health and poor wellbeing [11]. There is also a relatively higher proportion of sex workers [12]. In gay men and trans women, there is also a relatively high proportion of HIV/AIDS infection [13].

Despite these risk factors for increased COVID-19 infections, hospitalisations, and potential deaths, there is a lack of information on these rates amongst the UK LGBT population. There are several datasets that could be used to address this lack of data, but there are many issues that have thus far prevented this from taking place. The UK Office for National Statistics (ONS) has stated that it is “currently unable to report on deaths registered in England and Wales, including deaths involving COVID-19 by sexuality” (personal communication, Professor Sir Ian Diamond, Chief Statistician, ONS, 23 February 2021). Part of the reason for this is that sexual orientation and gender identity are not recorded in death certification, and partly because the UK Census 2011 did not include questions on sexual orientation and gender identity. The ONS COVID infection study does not ask questions relating to gender identify or sexual orientation. The Hospital Episode Statistics database also does not have sexual orientation or gender identity. The NHS Data Standard for Sexual Orientation Monitoring was adopted in 2019, but coverage is still very poor, possibly because “there is no need to collect data from every patient” [14]. There is no equivalent for gender identity yet.

UK datasets that do include sexual orientation include the ONS Annual Population Survey (APS), Health Survey for England (HSE), the Improving Access to Psychological Therapies (IAPT) cohort study, and the English General Practice Patient Survey (GPPS), but none of these have COVID-19 data. The Biobank cohort study has sexual behaviour data, but not sexual orientation. The ONS also holds data on civil partnerships and same-sex marriages, but this is a subset of around 15% of the minority sexual orientation population [3]. Understanding Society: The UK Household Longitudinal Study (UKHLS) records both COVID-19 data and sexual orientation, but the sample size is relatively small.

Datasets can be linked to the NHS database (Hospital Episode Statistics) to derive hospitalisations and deaths, as long as each of the linking databases have suitable identifiers, but linking datasets is statistically more complex than using a single database. It would be possible to link the APS, HSE, or IAPT databases this way, but not the GPPS. It would also be possible to link the Biobank or the partnerships data. None of these linkage projects have been done due to lack of staff time and other priorities of the ONS team (personal communication, J Tinsley, Head of Data, Health Analysis and Life Events Division, Public Policy Analysis, ONS; 9 June 2021).

The NHS workforce database also records sexual orientation as well as COVID-19 data, and, as the NHS employs around 1.5 million people [15], even though sexual orientation coverage may not be high, this could also be a very good source of data.

Large datasets that include gender identity are far fewer, and it is hoped that the addition of both sexual orientation and gender identity questions in the UK Census 2021 will help to alleviate this lack of data.

This study uses information from the UKHLS to derive the first information available on the rates of COVID-19 symptoms and positive COVID-19 tests, as well as other related information, by sexual orientation identity in the UK population.

## 2. Materials and Methods

### 2.1. Data and Materials

Sample: The data come from all seven waves of the UKHLS COVID-19 surveys [16,17]. These surveys were collected via the Web in April, May, June, July, September, and November 2020, as well as January 2021. Respondents to the COVID-19 surveys were respondents to wave 9 of the UKHLS, 2017–2019. The UKHLS has interviewed all adult household members from the sample annually since 2009. These COVID-19 surveys cover a variety of topics, including COVID-19 symptoms, testing, hospitalisation, childcare, key working status, furlough, mental health, and health behaviours. Approximately 17,800 individuals completed the April 2020 survey, and the number of responses decreased to just under 12,000 by January 2021.

### 2.2. Variables

Sexual Orientation: The UKHLS asked individuals about their sexual orientation in the main survey, as well as waves 3 (2011–2013) and 9 (2017–2019) of the survey. One standard question was used to assess sexual orientation, and responses included ‘heterosexual or straight’ (reference group), ‘gay or lesbian’, ‘bisexual’, ‘other’, and ‘prefer not to say’. We took information from both waves 3 and 9. If responses were not consistent, then the latest response was taken. Individuals who identified as ‘other’ or ‘prefer not to say’ were dropped from this analysis.

COVID-19 Symptoms, Testing, and Hospitalisation: At each month of the COVID-19 surveys, individuals were asked if they had experienced any COVID-19 symptoms, had been tested, or had been hospitalised due to COVID. In the April 2020 survey, individuals were asked if they had ever experienced symptoms, been tested, or were hospitalised, while, in the following surveys, respondents were asked if any of these had happened since the last COVID-19 survey. The reference group for these dichotomous variables was the ‘no’ response.

Covariates: The covariates included in the models were gender, age, highest educational qualification, ethnicity, diagnosed medical condition, and key worker status. Gender was a dichotomous variable, with men as the reference group. Highest educational qualification was a three-category variable. The responses were degree (i.e., university degree or higher; reference), A-level, or other higher qualifications (i.e., A-levels = exams taken at age 18 (year 13); other higher = teaching, nursing, or diploma certifications), and GCSE or lower (i.e., GCSE: General Certificate of Secondary Education, exams taken at age 16 (year 11); lower qualifications: Certificate of Secondary Education, skills certifications, apprenticeships, and clerical qualification). Ethnicity was a four-category variable, with responses of White British (reference), Black African or Caribbean, Indian, Pakistani or Bangladeshi, and other ethnicity. Diagnosed medical condition and key worker status were dichotomous variables, with ‘no’ as the reference group. Individuals were asked if they had ever been diagnosed with any of 17 possible chronic conditions. These included asthma, arthritis, diabetes, cancer, depression, high blood pressure, etc. At the first COVID-19 survey, individuals were asked if they had ever been diagnosed, and at each subsequent survey, they were asked if they had been diagnosed since their last completed survey. Key workers were defined by the UK government, and included individuals who worked in health and social care, education and childcare, key public services, local and national government, food services, public safety and national security, transportation, and utilities, communications, and financial services. Not all individuals who worked in these sectors were considered key workers, however, individuals should have been aware of whether they met the keyworker status requirements. Age was a continuous variable.

### 2.3. Statistics

COVID-19 symptoms, testing, and hospitalisation proportions were described by sexual orientation. Logistic regression models were run to determine whether COVID-19 symptoms or testing differed by sexual orientation. Models were not run for hospitalisations due to small numbers amongst sexual minorities. Outcomes were pooled across all months. Thus, the models tested if an individual had ever experienced COVID-19 symptoms or had been tested for COVID-19 at any month. Due to differences in reporting and experience of these outcomes by gender, interactions between sexual orientation and gender were conducted. The results from two models are presented; the first model includes all covariates, and the second model includes the gender and sexual orientation interaction term. Odds ratios are presented for the first model, and predicted probabilities given for the second model.

## 3. Results

### 3.1. Nature of the Sample

Compared to heterosexual individuals, gays and lesbians were more likely to have degrees, were less ethnically diverse, were younger, and were slightly more likely to have a diagnosed medical condition or be keyworkers (Table 1). Similarly, bisexual individuals were younger than heterosexual individuals, however, while a similar proportion had a degree, more bisexual individuals had obtained A-levels. Additionally, bisexual individuals were more ethnically diverse, were less likely to be a key worker, and were less likely to have a diagnosed medical condition. These patterns were similar by gender, with the exception of key workers. Lesbians were more likely to be key workers compared to either heterosexual or bisexual women. Gay and bisexual men were no more or less likely to be key workers compared to heterosexual men.

### 3.2. COVID-19 Results

The proportions of individuals who experienced COVID-19 symptoms, testing, or hospitalisation differed by sexual orientation (Table 2). A larger proportion of sexual minorities experienced symptoms compared to heterosexual individuals. While the number of symptoms experienced was similar between heterosexual and gay or lesbian respondents, bisexual individuals reported experiencing a greater number of symptoms. Similarly, gays and lesbians were no more or less likely to report having been tested, however, a larger proportion of bisexual individuals were tested. There was no apparent difference in the proportion of individuals who had been hospitalised by sexual orientation. Again, there were differences by gender. While bisexual men had similar proportions of individuals who had experienced COVID-19 symptoms, a larger proportion of bisexual women had experienced symptoms compared to either heterosexual women or lesbians. A larger proportion of lesbian and bisexual women experienced five or more symptoms compared to heterosexual women. While a larger proportion of gay men were hospitalised, a smaller proportion of lesbians were hospitalised compared to heterosexual men or women, respectively. Conversely, a smaller proportion of bisexual men were hospitalized, while a larger proportion of bisexual women were hospitalised compared to heterosexual men or women, gay men, or lesbians.

### 3.3. Regression Models

After adjusting for sociodemographic characteristics, gays, lesbians, and bisexual individuals were no more or less likely to have experienced COVID-19 symptoms compared to heterosexual individuals (Table 3). There were some differences by sociodemographic characteristics. For example, women, those of other ethnicity, key workers, and those with diagnosed medical conditions were more likely to have experienced symptoms. Individuals with lower levels of education or who were older were less likely to have experienced any symptoms. The inclusion of an interaction between sexual orientation and gender showed that there were no significant differences in experiences of COVID-19 symptoms by sexual orientation and gender.

Heterosexual females had a higher probability (23%) of having experienced COVID-19 symptoms than heterosexual males (18%). There was no statistical difference between gay men and lesbians or between bisexual males and females (Table 4). Additionally, there were no differences in the probability of having experienced COVID-19 symptoms among males of different sexual orientations or among females of different sexual orientations. For example, the probability of having experienced COVID-19 symptoms amongst females was 23% for heterosexual individuals, 20% for lesbians, and 28% for bisexual individuals, however, these were not significantly different. While some of these differences were approaching significance, none were at the 0.05 level, which may have been due to the sample sizes.

Similarly, there were no statistical differences in the likelihood of being tested for COVID-19 between gays or lesbians, bisexual individuals, and heterosexual individuals (Table 5). Similar to the symptoms model, women and key workers were more likely to have been tested. Individuals with lower than degree education or who were of Indian, Pakistani, or Bangladeshi ethnicity were less likely to have been tested.

Again, the interaction showed a difference in the probability of having been tested between heterosexual males (28%) and females (35%, Table 6). However, there were no statistical differences between males and females in either sexual minority group. Additionally, within males or females, there were no statistical differences in the probability of having been tested amongst sexual orientation groups. For example, the probability of having been tested was 28% for heterosexual males, 31% for gay males, and 36% for bisexual males. Some differences did approach significance, suggesting that differences may have been observed with larger sample sizes.

## 4. Discussion

### 4.1. Statement of Principal Findings

This ground-breaking study has shown that there are differences in the proportions of COVID-19 symptoms and likelihood of having a COVID-19 test by sexual orientation groups, but these differences mostly disappear when other characteristics, such as employment and ethnicity, are taken into account. So, despite the differences in the rates of adverse health behaviours and illnesses associated with higher rates of COVID-19 infections and hospitalizations, no apparent differences between groups were found. It is unclear whether this is due to the small sample sizes available in the UKHLS.

### 4.2. Strengths and Weaknesses of this Study

A major strength of this study is that it uses nationally representative data and that it is able to compare directly COVID-19 and health characteristics between sexual orientation groups without using data linkage. Linking population-based administrative information from two or more databases is very useful in providing detailed individual information from different sources for research purposes. In particular, it can answer questions requiring large sample sizes or detailed data on populations, such as sexual minority or gender minority status. However, there can be biases from linkage errors, or where records cannot be linked or are linked incorrectly, reducing the accuracy of the results [18]. Using a single database means no linkage errors, but is limited by the questions asked for that dataset. In this situation, the results are also limited by small sample sizes, so the ability to pick out small differences in COVID-19 symptoms and the probability of having a COVID-19 test will be unlikely. We were unable to analyse hospitalisation data due to the small sample sizes.

Another limitation is that sexual orientation is determined by a single question and does not take into account other aspects of sexual preference, such as behaviour or ideation. Furthermore, while the assessment was conducted twice, changes in sexual orientation were not included in the analysis, and only the last recorded sexual orientation was used. Another limitation is that it was not possible to analyse by gender identity, as the UKLHS does not include that attribute in the dataset. We chose not to include the ‘other’ and ‘prefer not to say’ groups, as the sexual orientation identity of people responding thus is unclear.

### 4.3. Comparison with Other Studies and Discussing Important Differences in Results

To date, there have been no other estimates published in the world on the incidence, hospitalization, or death rates in sexual and gender minority populations. A number of countries have been calling for this data, for example, in the US, where it is known that sexual minority individuals have a higher self-reported prevalence of several underlying health conditions associated with severe outcomes from COVID-19 than do heterosexual individuals [19].

### 4.4. Meaning of the Study: Possible Explanations

This study can only give a small amount of information on COVID-19 incidence by sexual orientation because of the small sample sizes available. However, one issue that can be taken from this study is that the lack of data on COVID-19 in LGBT populations mirrors the lack of health information by sexual orientation or gender identity in other conditions. For example, there is no good UK data on cancer by sexual orientation or gender identity, and sexual orientation was only added to the National Disease Registration Service (NDRS) core dataset in 2018, becoming mandatory in 2020, but reporting has been delayed by COVID-19 issues (personal communication, Andrew Murphy, Head of Cancer Datasets, NDRS, PHE, July 2020). It is unclear when they will add gender identity to this dataset. It has been thought that this lack of data is partly due to national policy discussions, which frame LGBT issues within an “it’s getting better” narrative, so there is a failure to examine what is actually happening [20]. 

The current UK Government Minister for Equalities stated that, “we have not found that LGBT groups specifically have been disproportionately affected” [21]. She has conflated a lack of data with the assumption of no effect. This lack of enquiry suggests an unwillingness to adequately acknowledge that there may well be health inequalities, and that efforts should be made to find and address them.

### 4.5. Implications for Clinicians, UK and International Policymakers, and Patients

Institutional homophobia is a relatively new concept, similar to institutional racism, and has been defined as: “The collective failure of an organisation to provide an appropriate and professional service to people because of their sexuality. It can be detected in processes, attitudes and behaviour which amount to discrimination through unwitting prejudice, ignorance, thoughtlessness and stereotyping” [22].

Official data collection that omits sexual orientation and gender identity is a subtle form of institutional homophobia/transphobia. The message appears to be that if these factors are not counted, populations with these attributes are not important. In 2017–2018, the UK government made great strides in promoting LGBT issues through their LGBT Action Plan [23], which included the development and uptake of sexual orientation and trans status monitoring in healthcare, as well as the recruitment of the UK’s first LGBT Health Advisor. Unfortunately, the Action Plan initiative, which would have delivered major advances, is no longer promoted by the current UK government [24].

### 4.6. Unanswered Questions and Future Research

It is known that men are more at risk of intensive care unit hospitalisations and deaths than women [25], despite equal infection rates from COVID-19. However, it is unclear why this is happening. Many theories have been suggested around lifestyle, access to healthcare, prevalence of pre-existing conditions, and physiological differences, such as prevalence of ACE2 receptors, sex hormone milieu, differential immune system strengths, and factors associated with the X chromosome [26]. Investigating the relative rates of COVID-19 hospitalisations and deaths in trans men and trans women may help to provide further evidence for some of these factors. Lesbians may have higher testosterone levels [27], and it is unclear if this is associated with better or worse COVID-19 outcomes, including long COVID syndrome.

Although we have not been able to show higher infection rates in gay men from this study, which would have been expected from the data from the National Survey of Sexual attitudes and Lifestyles (NATSAL) study [5] mentioned previously, the small sample size in the UKHLS means that the principal questions remain around infection rates, symptom severity, hospitalisations, intensive care unit admissions, and deaths from COVID-19 in UK sexual and gender minority populations. These should be supplied as soon as possible by whatever means available to researchers, so that suitable action can be taken as necessary to minimise future risks to people within the sexual and gender minority communities.

Finally, the reasons why sexual and gender minorities have been omitted from all of the major data collection activities in the UK on COVID-19 needs to be explored, and action needs to be taken to ensure that this does not happen in the future.

## 5. Conclusions

There are a number of reasons why incidence, hospitalisations, and death rates from COVID-19 in sexual and gender minorities might be higher than in the majority population. This study uses information from the UKHLS to derive the first information available on the rates of COVID-19 symptoms and positive COVID-19 tests by sexual orientation identity in the UK population. The proportions of gay/lesbian and particularly bisexual individuals reporting COVID-19 symptoms and taking COVID-19 tests was higher than heterosexual individuals, but most of the results were not statistically significant in the regression models that took other characteristics into account, probably because of small sample sizes. Future studies must ensure better representation of minority group data.

## Figures and Tables

**Table 1 healthcare-09-00937-t001:** Individual sociodemographic characteristics by sexual orientation and gender *.

Sociodemographic Characteristics	Total			Men			Women		
	Heterosexual (*n* = 17,400)	Gay or Lesbian (*n* = 324)	Bisexual (*n* = 293)	Heterosexual (*n* = 7265)	Gay (*n* = 193)	Bisexual (*n* = 93)	Heterosexual (*n* = 10,132)	Lesbian (*n* = 130)	Bisexual (*n* = 198)
**Highest educational qualification**									
Degree	5980 (28)	159 (49)	113 (28)	2172 (29)	74 (49)	36 (36)	1997 (27)	51 (51)	61 (26)
A-level or similar	5771 (34)	88 (31)	100 (46)	1995 (35)	47 (31)	24 (46)	2480 (34)	28 (32)	55 (49)
GCSE or lower	4695 (38)	50 (20)	55 (26)	1606 (36)	27 (21)	12 (18)	2918 (39)	17 (17)	34 (25)
**Ethnicity**									
White BRITISH	14,619 (88)	293 (94)	223 (82)	5156 (87)	137 (93)	54 (80)	6926 (88)	96 (97)	128 (82)
Black African/Caribbean	454 (6)	2 (0)	2 (1)	110 (2)	1 (0)	0 (0)	234 (2)	1 (0)	2 (1)
Indian/Pakistani/Bangladeshi	1103 (4)	3 (0)	23 (5)	386 (5)	2 (1)	12 (13)	485 (4)	0 (0)	9 (1)
Other ethnicity	1090 (6)	25 (5)	40 (13)	346 (6)	18 (6)	11 (7)	564 (6)	23 (3)	23 (16)
**Keyworker**									
No	11,851 (73)	201 (71)	200 (79)	4434 (76)	113 (78)	58 (80)	5178 (71)	54 (56)	111 (77)
Yes	4588 (27)	107 (29)	68 (21)	1306 (24)	38 (22)	16 (20)	2372 (29)	41 (44)	40 (23)
**Diagnosed medical condition**									
No	6938 (40)	113 (36)	128 (47)	2369 (42)	62 (37)	33 (48)	(38)	32 (35)	77 (44)
Yes	10,462 (60)	211 (64)	165 (53)	3676 (58)	96 (64)	44 (52)	5026 (62)	69 (65)	87 (56)
**Mean age**	49.98	44.54	33.02	49.87	45.06	36.62	50.09	43.42	31.12

* Raw numbers and weighted percentages in parentheses; weighted means.

**Table 2 healthcare-09-00937-t002:** COVID-19 symptoms and testing amongst individuals by sexual orientation and gender *.

COVID-19 Symptoms	Total			Men			Women		
	Heterosexual (*n* = 14,310)	Gay (*n* = 260)	Bisexual (*n* = 243)	Heterosexual (*n* = 6045)	Gay (*n* = 158)	Bisexual (*n* = 77)	Heterosexual (*n* = 8263)	Lesbian (*n* = 101)	Bisexual (*n* = 164)
**Has had COVID symptoms**									
No	11,413 (80)	188 (75)	162 (64)	4954 (83)	110 (75)	54 (78)	6457 (79)	75 (75)	107 (61)
Yes	2897 (19)	72 (25)	81 (35)	1091 (17)	46 (25)	23 (22)	1806 (21)	26 (25)	57 (39)
**Number of COVID symptoms**									
No symptoms	55 (2)	1 (0)	2 (2)	17 (1)	1 (0)	0 (0)	38 (2)	0 (0)	2 (4)
One symptom	215 (7)	1 (2)	4 (2)	84 (8)	1 (4)	2 (2)	131 (6)	0 (0)	2 (2)
Two symptoms	386 (10)	8 (14)	6 (4)	126 (11)	8 (21)	1 (1)	160 (9)	0 (0)	5 (5)
Three symptoms	379 (14)	10 (15)	5 (5)	178 (16)	6 (17)	3 (16)	201 (12)	4 (11)	2 (2)
Four symptoms	378(14)	8 (14)	5 (12)	151 (17)	5 (17)	1 (3)	227 (12)	3 (8)	3 (3)
Five or more symptoms	1584 (54)	44 (55)	59 (76)	536 (48)	25 (42)	16 (78)	1048 (59)	19 (81)	43 (85)
**Tested for Coronavirus**									
No	10,044 (72)	165 (70)	162 (65)	4447 (76)	96 (68)	58 (82)	5596 (69)	69( 76)	103 (58)
Yes	4265 (28)	94 (30)	81 (35)	1597 (24)	62 (32)	19 (18)	2667 (31)	31 (24)	61 (42)
**Hospitalised due to COVID symptoms**									
No	13,675 (96)	250 (97)	229 (96)	5820 (97)	152 (97)	74 (99)	7853 (95)	97( 97)	153 (94)
Yes	604 (4)	10 (3)	11 (4)	216 (3)	6 (3)	2 (1)	388 (5)	4 (3)	9 (6)

* Raw numbers and weighted percentages in parentheses.

**Table 3 healthcare-09-00937-t003:** Odds ratios of sexual orientation as a predictor for COVID-19 symptoms.

**Sociodemographic Characteristics**	**Model 1**	
	Odds Ratio	95% Confidence Interval
**Sexual orientation (Ref = Heterosexual)**		
Gay/Lesbian	0.98	(0.65, 1.47)
Bisexual	1.62	(0.79, 3.33)
**Gender (Ref = Male)**		
Female	1.25	(1.07, 1.47)
**Age**	0.98	(0.98, 0.99)
**Educational attainment (Ref = Degree)**		
A-level or similar	0.75	(0.63, 0.89)
GCSE or lower	0.66	(0.53, 0.83)
**Ethnicity (Ref = White British)**		
Black African/Caribbean	1.28	(0.72, 2.28)
Indian/Pakistani/Bangladeshi	0.82	(0.57, 1.16)
Other Ethnicity	1.66	(1.14, 2.42)
**Keyworker (Ref = No)**		
Yes	1.53	(1.29, 1.82)
**Ever diagnosed with medical condition (Ref = No)**		
Yes	1.18	(1.00, 1.39)

**Table 4 healthcare-09-00937-t004:** Marginal probabilities of experiencing COVID-19 symptoms by sexual orientation and gender.

	Sexual Orientation		
Gender	Heterosexual	Gay/Lesbian	Bisexual
Male	0.18	0.19	0.35
Female	0.23	0.20	0.28
Contrast between Gender	*		

* = Difference between males and females.

**Table 5 healthcare-09-00937-t005:** Odds ratios of sexual orientation as a predictor for COVID-19 testing.

**Sociodemographic Characteristics**	**Model 1**	
	Odds Ratio	95% Confidence Interval
**Sexual orientation (Ref = Heterosexual)**		
Gay/Lesbian	0.93	(0.61, 1.41)
Bisexual	1.25	(0.60, 2.59)
**Gender (Ref = Male)**		
Female	1.29	(1.13, 1.48)
**Age**	1.00	(0.99, 1.00)
**Educational attainment (Ref = Degree)**		
A-level or similar	0.82	(0.71, 0.95)
GCSE or lower	0.78	(0.65, 0.94)
**Ethnicity (Ref = White British)**		
Black African/Caribbean	0.73	(0.42, 1.27)
Indian/Pakistani/Bangladeshi	0.50	(0.45, 0.71)
Other Ethnicity	1.37	(0.97, 1.93)
**Keyworker (Ref = No)**		
Yes	2.31	(1.99, 2.69)
**Ever diagnosed with medical condition (Ref = No)**		
Yes	1.12	(0.98, 1.29)

**Table 6 healthcare-09-00937-t006:** Marginal probabilities of having been tested for COVID-19 by sexual orientation and gender.

	Sexual Orientation		
Gender	Heterosexual	Gay/Lesbian	Bisexual
Male	0.28	0.31	0.36
Female	0.35	0.22	0.38
**Contrast between Gender**	*		

* = Difference between males and females.

## Data Availability

UKHLS data are accessible via the UKDS website: https://www.understandingsociety.ac.uk/documentation/access-data, accessed on 5 May 2021.

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
