# Peer review of "Sexual Orientation and the Incidence of COVID-19: Evidence from Understanding Society in the UK Longitudinal Household Study"

_healthcare, 2021, doi:10.3390/healthcare9080937_

Round 1
Reviewer 1 Report
This paper is a unique topic with regards to COVID-19. And the introduction section clearly describes the current situation and necessity.
A bit confusing to read in the 'Materials and Methods' section. It would be better to divide the sub-section into Data and materials, variables, statistics, etc.
In variable names, ‘degree’, ‘A-level’ and ‘GCSE’ are difficult to understand in other countries and do not know what they mean. Define variables in detail for international readers. What is the definition of Keyworker? What is a diagnosed medical condition? Please elaborate on the definition of the main variable.
There seems to be no reason not to indicate the subject number in Tables 1 and 2. Please indicate.
“The lack of data 309 is very worrying, and future data collection must be improved.” This expression is slightly inappropriate to describe in the “Conclusion section”. And it has already been mentioned several times as a limitation. Please delete it
Reviewer 2 Report
Hello,
Thank you for the opportunity to review this manuscript. The paper is well-written and the topic is timely. My main suggestions has to do with the framing of the paper. The results indicate that "there was no apparent difference in the proportion of individuals who had been hospitalised by sexual orientation." Instead, gender, education, and ethnicity (to a lesser extent) were more reliable predictors of differences in symptoms and hospitalizations. To me, this is a positive result--why would we expect gay men, lesbian women, and bisexual persons to have worse COVID-19 outcomes, when just looking at sexual orientation alone? Is there something biologically or genetically different in homosexual persons? Persons of sexual minority IDs certainly face discrimination in healthcare, but that is not what this study is measuring. I encourage you to slightly adjust your framing so that it's clear that you are simply describing the data and not predicting differences. Also, I suggest you use the term "bisexual persons" not "bisexuals" in the paper.
